# Essential Thrombocythemia and Ischemic Stroke: A Case Series of Five *JAK2*-Positive Patients

**DOI:** 10.3390/medicina59071300

**Published:** 2023-07-14

**Authors:** Byong-Kyu Kim, Kyung Yoon Eah, Jin-Mo Park

**Affiliations:** 1Division of Cardiology, Department of Internal Medicine, Dongguk University Gyeongju Hospital, Dongguk University College of Medicine, Gyeongju 38067, Republic of Korea; bleumatin@dongguk.ac.kr; 2Department of Neurology, Dongguk University Gyeongju Hospital, Dongguk University College of Medicine, Gyeongju 38067, Republic of Korea; neu727@dongguk.ac.kr

**Keywords:** essential thrombosis, ischemic stroke, *JAK2*, recurrent stroke, cytoreductive therapy

## Abstract

*Background and Objectives*: Essential thrombocythemia (ET) is a chronic myeloproliferative neoplasm characterized by elevated platelet counts and an increased risk of thrombotic events, including ischemic strokes. Materials and *Methods*: We conducted a retrospective analysis of data from consecutive ischemic stroke patients with ET between March 2014 and February 2023. *Results*: This case series describes the clinical presentation, radiological features, and management of five patients with ET-associated ischemic strokes, all harboring the *JAK2* mutation. The diverse radiological findings suggest that both large and small vessel diseases may be influenced by the prothrombotic state induced by ET. A significant elevation in platelet count was observed to correlate with the emergence of new acute infarctions in some cases. *Conclusions*: The study highlights combined use of antiplatelet and cytoreductive therapy in preventing secondary stroke events in patients with ET and *JAK2* mutations. The heterogeneity of stroke patterns in this population necessitates a comprehensive understanding of the underlying pathophysiological mechanisms and tailored therapeutic approaches.

## 1. Introduction

Essential thrombocythemia (ET) is a rare, chronic myeloproliferative neoplasm characterized by the overproduction of platelets [1]. Despite its rarity, ET holds significant clinical relevance due to the potential for thrombotic and hemorrhagic events. In the global context, ET has an estimated incidence of 1.2 to 3.0 per 100,000 individuals per year, and a prevalence of approximately 30 per 100,000 individuals, which highlights the global burden of this condition [1,2]. In South Korea, the situation is similar, with the most recent crude annual incidence of ET being 0.84 per 100,000 individuals, and the crude prevalence rate being 7.10 per 100,000 individuals [3]. The majority of patients with ET have a normal life expectancy, adding complexity to the management and study of this disease [2,3,4]. The diagnosis of ET is established based on the World Health Organization (WHO) criteria, which include elevated platelet counts, bone marrow biopsy findings, and the presence of a *JAK2*, *CALR*, or *MPL* mutation [5]. The *JAK2* V617F mutation, in particular, is a key molecular driver of the disease and has been identified in approximately 50–60% of ET cases [6]. The *JAK2* mutation contributes significantly to the pathogenesis of ET by leading to constitutive activation of the Janus kinase-signal transducer and activator of transcription signaling pathway. This results in unregulated platelet production, creating a hypercoagulable state that increases the risk of thrombotic complications [7].

ET can manifest in varying clinical presentations, ranging from asymptomatic conditions to severe complications. Beyond the thrombotic complications, such as ischemic strokes, patients with ET can experience significant hemorrhagic events and cardiovascular abnormalities [8]. Ischemic strokes represent a major concern due to the potential for long-term morbidity and mortality [7,8,9]. Ischemic strokes, in particular, can lead to severe complications, including long-term disability, cognitive impairment, and reduced quality of life [10].

The overarching aim of this study is to enhance our understanding of ET, particularly in patients harboring the *JAK2* mutation and experiencing ET-associated ischemic strokes. We will present a detailed analysis of five such patients, encompassing their clinical presentations, diagnostic workup, and management strategies. By doing so, we aim to fill existing gaps in the literature, contribute to improved patient care, and advance our collective knowledge on this critical topic.

## 2. Methods

We conducted a retrospective analysis of data from consecutive ischemic stroke patients with definitive ET treated at the department of neurology in our hospital between March 2014 and February 2023. ET was diagnosed according to the World Health Organization criteria, which include a sustained platelet count ≥450,000/μL, a bone marrow biopsy specimen showing increased numbers of enlarged, mature megakaryocytes, not meeting the criteria for other chronic myeloid neoplasms, and the *JAK2* 617 V mutation, or no evidence of reactive thrombocytosis [5].

The data were collected from the electronic medical records (EMRs) at our hospital. Pertinent data on clinical characteristics, location of ischemic stroke, and laboratory data including platelet count, hemoglobin level, leukocyte count, and *JAK2* V617F mutation status were extracted. We also assessed the patients’ outcomes and the treatments they received as part of our review process. The study was approved by the Institutional Review Board of Dongguk University Gyeongju Hospital (110757-202347-HR-04-02), and informed consents were waived for this study given its retrospective design.

## 3. Results

Out of 961 patients admitted to the neurology department during the period from March 2014 to February 2023, we identified five cases (0.52%). These cases, comprising four women and one man, represent a specific subset of individuals diagnosed with ischemic stroke and *JAK2* mutation-associated ET. The patients had a mean age of 71.8 ± 8.6 years (range, 63–85 years) and presented with various medical histories and stroke symptoms (Table 1). The previous medical histories relevant to stroke risk factors included hypertension in four patients, diabetes mellitus in none, and hyperlipidemia in one patient. One patient (Case 5) had a history of ischemic stroke and had been treated with antiplatelet drugs prior to the current stroke event. The remaining patients had no previous history of stroke, transient ischemic attack (TIA), or recognized neurological symptoms.

The mean platelet count at the time of stroke was 811,600 ± 112,528/μL (range, 643,000–949,000/μL). *JAK2* mutations were confirmed in all cases. Bone marrow biopsies were performed in all cases, revealing various findings. All patients received combined antiplatelet and cytoreductive therapy following the stroke. The therapeutic goal for platelet count is to maintain it below 600,000/μL [11]. A hematologist performed monitoring of platelet counts (every 1–2 months) and selected the appropriate cytoreductive therapy. The status of patients was generally good, with most achieving a modified Rankin Scale score of 0–1 at discharge. The follow-up periods for the cases were as follows: 5.5 years for Case 1, 4.5 years for Case 2, 4 years and 3 months for Case 3, 2 years and 9 months for Case 4, and 3 months for Case 5, with a mean follow-up period of 3.5 ± 1.9 years. The patients remained stable during the follow-up period, with no recurrences reported, although the follow-up period for Case 5 was too short for definitive conclusions.

### 3.1. MRI Findings

MRI findings in this case series demonstrated a diverse range of acute and chronic infarctions, lacunar lesions, white matter hyperintensities, and small vessel disease. In some instances, patients exhibited acute lesions and old leukoaraiosis, despite the absence of prior history of stroke or recognized neurological symptoms. Furthermore, small vessel occlusions and large vessel atherosclerosis were identified. Lesions were observed in both the anterior and posterior circulation areas, with certain cases exhibiting a widespread distribution of lesions throughout the brain.

### 3.2. Detailed Case Presentations

Case 1: A 64-year-old female presented with weakness in her left extremity. MRI findings revealed multiple lesions within the territory of the right MCA, specifically in the temporal and frontoparietal lobes, which suggested embolic infarction (Figure 1A). However, no relevant intracranial and extracranial vascular abnormalities were detected. Furthermore, cardiac evaluations, including transthoracic echocardiogram (TTE) and Holter monitoring, failed to identify any potential embolic sources. Additionally, she had several lacunes, indicative of old small infarct lesions, in the right frontoparietal subcortical area. Initial platelet count was 949,000/μL. Bone marrow examination revealed mild hypercellularity and marked megakaryocytic hyperplasia. The patient received cytoreductive therapy (ruxolitinib) and took antiplatelet medication (from clopidogrel (4 years) to cilostazol (18 months)). Notably, her hospital stay was prolonged (60 days) due to an accidental fall leading to an orthopedic intervention and subsequent pneumonia. Despite these complications, she was discharged with good recovery from her stroke. During the 5.5-year follow-up, no recurrence occurred, and her condition remained stable.

Case 2: An 85-year-old female with a past medical history of hypertension and hyperlipidemia presented with persistent vertigo. Imaging demonstrated a small acute infarction in the left occipital lobe and several lacunes in the right cerebellar hemisphere, both basal ganglia, and right corona radiata (Figure 1B). Initial platelet count was 802,000/μL. Bone marrow biopsy showed normocellular to hypercellular marrow with increased and enlarged megakaryocytes. She received cytoreductive therapy (hydroxyurea) and clopidogrel for secondary stroke prevention, and follow-up MRI after 4.5 years showed no new lesions or interval changes.

Case 3: A 67-year-old male with hypertension presented with acute vertigo and ambulatory difficulties due to impaired balance. MRI revealed acute embolic infarctions in the left lateral medulla, both cerebellar hemispheres, and left temporo-occipital lobe (Figure 1C). MRA displayed occlusion of the left VA V4 segment (Figure 1C). The patient’s initial platelet count was 829,000/μL. The patient’s bone marrow showed hypercellularity and increased megakaryocytes. Despite a low initial NIHSS score, the patient required extended rehabilitation due to severe gait disturbance from imbalance, and sensory deficits in the left hemiface were not fully recovered. Despite persistent sensory deficits, the patient remained stable for over 4 years following anagrelide, hydroxyurea, and clopidogrel treatment. A follow-up MRI and MRA after 4 years and 3 months showed no new lesions or interval changes.

Case 4: An 80-year-old female presented with dizziness, and ambulatory difficulties. She had a past medical history of hypertension. MRI findings revealed acute infarction in the right temporoparietal deep white matter, subacute infarction in the right frontoparietal deep white matter and cystic encephalomalacia in the right temporo-occipital lobe (Figure 1D). The patient’s initial platelet count was 729,000/μL. Bone marrow biopsy showed slightly hypercellular marrow for her age and increased megakaryocytes. Following cytoreductive treatment with hydroxyurea and antiplatelet therapy with clopidogrel, her condition remained stable for 2 years and 9 months.

Case 5: A 63-year-old female with a history of cerebral infarction, myocardial infarction and essential thrombocytosis, had discontinued her medication for approximately one month. During this period, her platelet count increased to 600,000/μL. She presented with a two-week history of right upper and lower extremity weakness (Grade 4+). MRI revealed a tiny acute infarction in the left posterior periventricular white matter and high parietal lobe, as well as cystic encephalomalacia in the right temporo-occipital lobe and focal cystic encephalomalacia in the right frontal lobe (Figure 1E). The patient’s bone marrow revealed hypercellularity and hyperplasia of megakaryocytes. She received combined antiplatelet (aspirin and clopidogrel) and cytoreductive therapy (hydroxyurea) after stroke and remained stable after 3 months; however, ongoing monitoring and adherence to medication were needed.

## 4. Discussion

The diverse radiological features observed in this series highlight the complexity of stroke manifestations in patients with ET, offering insights into their management. Both large and small vessel diseases were present, suggesting that the prothrombotic state induced by ET could influence a wide array of vascular territories. This interpretation aligns with the findings of recent research emphasizing the heterogeneous nature of vascular involvement in ET-associated strokes [12,13]. Our observations of mixed vascular disease, as seen in cases 1, 2, and 5, underscore the extent of this complexity, with large and small vessel diseases co-existing in the same patients. However, it is also important to recognize that our patients may have other risk factors contributing to large vessel disease, thereby pointing towards a multifaceted interplay between ET and additional risk factors [14]. This interplay further underscores the critical need for individualized patient assessment and personalized therapeutic strategies. Interestingly, infratentorial infarctions were uncommonly reported. In earlier studies, such an event was found in merely one out of ten patients in one study [12], and strikingly, none were reported among eleven patients in another study [13]. In our series, however, we identified infratentorial lesions in two out of five patients (case 2 and 3). This notable finding further exemplifies the wide spectrum of stroke presentations associated with ET and underscores the need for more extensive exploration.

Furthermore, TIA is indeed a common manifestation of ET [15]. However, in our case series, neither TIA symptoms nor recognized neurological symptoms were noted in any of the five patients. Yet, on MRI, old or subacute lesions were evident in cases 1 and 4. This suggests that in ET, small-sized thromboembolic lesions may present as unrecognized lacunar infarctions. Conversely, it suggests the necessity for routine imaging evaluations to consider the possibility of silent infarctions even in ET patients without confirmed stroke. This strategy appears particularly relevant for ET patients with additional risk factors for stroke.

Despite imaging suggestive of embolic infarction, with features such as involvement of multiple vessel territories or small-sized cortical/subcortical lesions, it was challenging to definitively identify an embolic source through vascular and cardiac evaluations. The absence of transesophageal echocardiograms to potentially reveal cardiac or atrial thrombi represents a limitation of this study. In the context of ET, it is theoretically more common for thrombotic strokes to occur, as an overproduction of platelets can lead to clot formation within the brain’s blood vessels. Nonetheless, it is also feasible for a clot to develop elsewhere and migrate to the brain, causing an embolic stroke [16].

According to current guidelines, in cases where a *JAK2* mutation is confirmed and arterial thrombotic events such as stroke have occurred, treatment with hydroxyurea in combination with twice-daily aspirin is recommended [17,18]. However, in these case studies, clopidogrel was used instead of aspirin, diverging from the guidelines. This was due to the fact that ET was diagnosed after the initiation of clopidogrel for stroke, leading to the decision not to switch from clopidogrel to aspirin.

Case 5 exhibited a new acute infarction in the left posterior periventricular white matter and high parietal lobe after discontinuing essential thrombocytosis medication, resulting in an increased platelet count of 600,000/μL. This observation underscores a potential association between platelet count and stroke risk in ET patients. However, it is worth noting that findings from other study report no correlation between platelet counts and the type or occurrence of stroke [13]. The predominance of supratentorial lacunar infarcts and chronic white matter lesions was noted in this study, with the majority of patients undergoing antiplatelet or cytoreductive therapy. This raises the possibility that the use of antiplatelet or cytoreductive therapy may be more closely related to the risk of stroke recurrence than the increase in platelet count alone.

In our cohort of five patients, of whom one has been followed up for approximately three months and may be premature to assess, no stroke recurrence was observed during a mean follow-up period of 3.5 ± 1.9 years, suggesting a generally favorable prognosis. However, this observation somewhat differs from a previous study which reported recurrent stroke in three out of ten ET patients [12]. Unfortunately, due to lack of detailed information on these recurrent cases, such as changes in platelet count or medication compliance, a direct comparison is challenging. Discrepancies in findings may be due to differences in patient populations, treatment regimens, or follow-up durations, indicating that more research is needed to fully elucidate the factors influencing stroke recurrence in ET patients.

Our study has several limitations that need to be acknowledged. The small sample size and the retrospective design may limit the generalizability of our findings. Furthermore, the potential for selection bias cannot be discounted. Additionally, we did not perform transesophageal echocardiograms, which could have potentially revealed cardiac or atrial thrombi. Despite these limitations, our study provides valuable insights into the complex and varied radiological features of strokes in patients with ET. Future larger-scale, prospective studies could build upon our findings to further our understanding of ET-associated strokes, guiding the development of more effective, patient-specific therapeutic approaches.

## 5. Conclusions

Effectively managing and closely monitoring patients with ET are imperative for reducing stroke risk. Regular platelet count assessments can help detect changes in disease status, facilitating timely treatment adjustments to minimize stroke risk. Our findings emphasize the importance of medication adherence and the crucial role of combined antiplatelet and cytoreductive therapy in preventing secondary stroke events in patients with ET. The heterogeneity of stroke patterns in this population necessitates a comprehensive understanding of the underlying pathophysiological mechanisms and tailored therapeutic approaches.

## Figures and Tables

**Figure 1 medicina-59-01300-f001:**
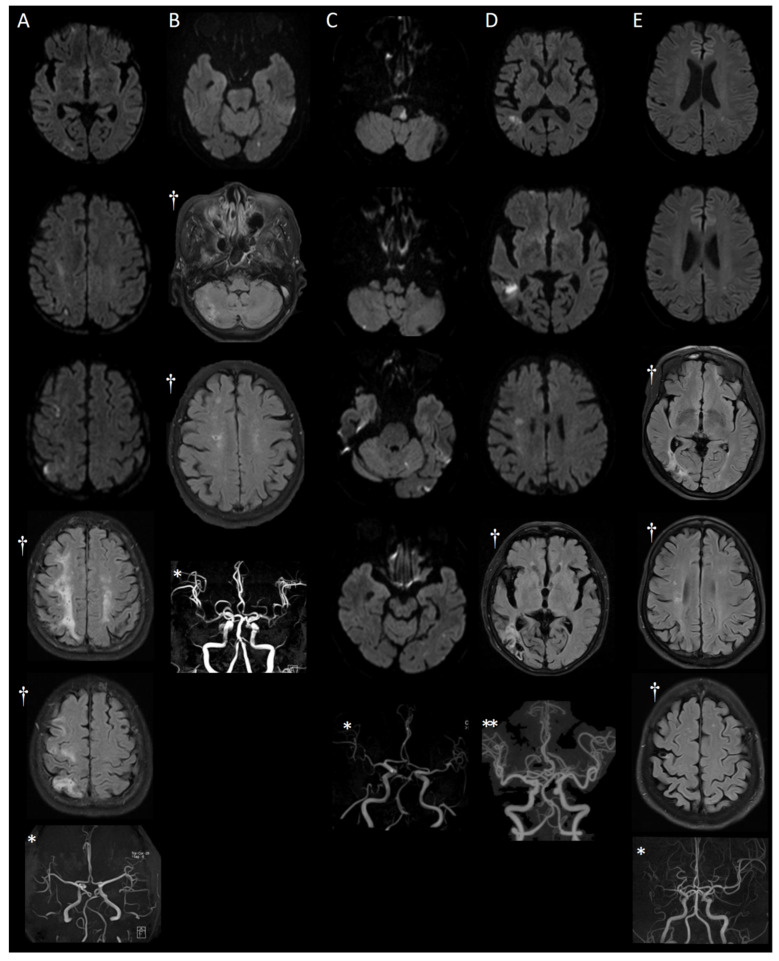
MRI findings of the five cases. (**A**) multiple embolic acute infarctions in the right temporal and frontoparietal lobes, and lacunes in the right frontoparietal subcortical area. (**B**) small acute infarction in the left occipital lobe and multiple lacunes in the right cerebellar hemisphere, basal ganglia, and right corona radiata. (**C**) acute embolic infarctions in the left lateral medulla, both cerebellar hemispheres, and left temporo-occipital lobe, along with occlusion of the left VA V4 segment on MRA. (**D**) acute infarction in the right temporoparietal deep white matter, subacute infarction in the right frontoparietal deep white matter, and cystic encephalomalacia in the right temporo-occipital lobe. (**E**) tiny acute infarction in the posterior periventricular white matter and high parietal lobe on the left side, cystic encephalomalacia in the right temporo-occipital lobe and focal cystic encephalomalacia in the right frontal lobe. † represent FLAIR, those marked with * represent TOF MRA, and those marked with ** indicate CTA. The unmarked images are DWI.

**Table 1 medicina-59-01300-t001:** Clinical characteristics of ET cases.

Case	1	2	3	4	5
Gender	Female	Female	Male	Female	Female
Age	64	85	67	80	63
Presenting symptom	Left hemiparesis	Dizziness	Dizziness and impaired balance	Dizziness and gait disturbance	Right hemiparesis
Risk factors					
HTN	-	+	+	+	+
DM	-	-	-		-
hyperlipidemia	+ * (LDL 86)	+	+ * (LDL 111)	+ * (LDL 98)	+
Smoking	-	-	-	-	-
B-MRA	Unremarkable	Mild luminal irregularities in both MCA bifurcations, Focal stenosis in both VAs V4 segment	Occlusion of left VA V4 segment	Unremarkable	Steno-occlusive lesion of right MCA
Proximal vessel	Carotid USG (isoechoic plaque in both carotid bulb and right ICA, Calcified plaque in right carotid bulb and right ICA)	None	CTA (steno-occlusion of left VA V3–V4 segment)	CTA (mild focal stenosis with calcified plaque at both carotid bulb) Carotid USG (Hyperechoic plaque in Rt carotid bulb to ICA)	CTA (No remarkable finding in extracranial both carotid artery system)
TTE	Normal LV function, diastolic dysfunction grade I, No intracardiac thrombus	Concentric LVH with normal LV systolic function, Diastolic dysfunction grade I, No intracardiac thrombus	Mild concentric LVH and normal LV systolic function, diastolic dysfunction grade I, No intracardiac thrombus	Moderate AS and mild AR, Normal LV systolic function, Enlarged LA, Diastolic dysfunction grade I, No intracardiac thrombus	Concentric LVH with normal LV systolic function, Enlarged LA, Diastolic dysfunction grade I, No intracardiac thrombus
Holter	none	Frequent APCs Frequent VPCs	Rare APCs	Frequent APCs Rare VPCs	none
Antiplatelet	clopidogrel → cilostazol	clopidogrel	clopidogrel	clopidogrel	aspirin + clopidogrel
Cytoreductive therapy	ruxolitinib	hydroxyurea	hydroxyurea	hydroxyurea	hydroxyurea
NIHSS, initial	4	1	4	1	2
mRS initial	3	2	4	2	1
Admission duration (days)	60	7	68	6	9
NIHSS, at discharge	1	0	1	0	0
mRS at discharge	1	1	1	0	0

HTN, Hypertension; DM, Diabetes Mellitus; B-MRA, Brain Magnetic Resonance Angiography; LDL, Low-Density Lipoprotein; USG, Ultrasonography; CTA, Computed Tomography Angiography; ICA, Internal Carotid Artery; TTE, Transthoracic Echocardiogram; LV, Left Ventricle; LVH, Left Ventricular Hypertrophy; AS, Aortic Stenosis; AR, Aortic Regurgitation; APC, Atrial Premature Complex; VPC, Ventricular Premature Complex; NIHSS, National Institutes of Health Stroke Scale. *, newly diagnosed risk factor after stroke.

## Data Availability

The data that support the findings of this study are available from the corresponding author upon reasonable request.

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
