# Peer review of "Essential Thrombocythemia and Ischemic Stroke: A Case Series of Five JAK2-Positive Patients"

_medicina, 2023, doi:10.3390/medicina59071300_

Round 1

Reviewer 1 Report

The study performed by authors “Essential Thrombocythemia and Ischemic Stroke: A Case Series of Five JAK2-Positive Patients”, This work showed some interesting information. there are some minor points to be further addressed. Mostly linked to the research topic and experimental results. However, the manuscript needs minor revision and following questions/points should be further clarified

Some questions and comments as follow:

Introduction:
While the introduction provides valuable background information, there are a few potential defects or limitations:

 1. Lack of context: The introduction does not provide sufficient contextual information about the prevalence or significance of ET. It would be helpful to include details about the rarity of ET and its impact on the population to better understand the importance of studying ET-associated ischemic strokes.

 2. Limited scope: The introduction focuses solely on ET and its association with ischemic strokes. It does not discuss other potential clinical manifestations or complications of ET, such as hemorrhagic events or cardiovascular abnormalities. Expanding the scope to include a broader overview of ET-related complications could provide a more comprehensive context for the study.

 3. Insufficient rationale: While the introduction states that the study aims to provide a detailed analysis of five patients with ET-associated ischemic strokes, it does not clearly explain why this analysis is necessary or how it will contribute to the existing knowledge in the field. Adding a clear rationale for conducting this specific study and highlighting any gaps in the current literature would enhance the introduction.

4. Lack of objectives: The introduction does not explicitly state the objectives or research questions of the study. Including clear objectives would help readers understand the purpose and focus of the research.

Methods:
5. The methods do not provide specific information about data collection procedures, the process of data extraction, how missing data were handled, and any statistical analyses performed. Including these details would enhance the transparency and reproducibility of the study.

Results:
6. What are the long-term recurrence rates of ischemic stroke in patients with JAK2 mutation-associated essential thrombocythemia (ET) and how do they compare to other patient populations?

 7. How does the combined antiplatelet and cytoreductive therapy influence outcomes and the maintenance of platelet counts below 600,000/μL in patients with JAK2 mutation-associated essential thrombocythemia (ET) and ischemic stroke.

Discussion:
8. The discussion does not compare the outcomes or characteristics of the study's patients to a control group or other populations. A comparison could provide insights into the uniqueness or similarities of the observed findings compared to patients without ET or different etiologies of ischemic stroke.

9. The discussion briefly explains the use of clopidogrel instead of aspirin for secondary stroke prevention in patients with confirmed JAK2 mutation and arterial thrombotic events. However, it does not discuss the rationale behind this decision or consider other possible treatment options. A more comprehensive analysis of the various therapeutic approaches available would provide a broader perspective on the management of ET-associated ischemic strokes.

 Conclusion:

Why doesn't the article have a conclusion section?

Minor editing of English language required

Author Response

We appreciate the time and effort the reviewers have taken to evaluate our manuscript. Their comments and suggestions have been invaluable in improving the quality and clarity of our work. Please find below our point-by-point response to each comment and concern raised.

Reviewer 1:

Introduction:

While the introduction provides valuable background information, there are a few potential defects or limitations:

  1. Lack of context: The introduction does not provide sufficient contextual information about the prevalence or significance of ET. It would be helpful to include details about the rarity of ET and its impact on the population to better understand the importance of studying ET-associated ischemic strokes.

> We appreciate your comment about the need to provide more context on the prevalence or significance of ET. To address this, we plan to include more details about the rarity of ET and its potential impact on patient populations to underscore the importance of studying ET-associated ischemic strokes.

  1. Limited scope: The introduction focuses solely on ET and its association with ischemic strokes. It does not discuss other potential clinical manifestations or complications of ET, such as hemorrhagic events or cardiovascular abnormalities. Expanding the scope to include a broader overview of ET-related complications could provide a more comprehensive context for the study.

> We acknowledge the importance of discussing other potential clinical manifestations or complications of ET apart from ischemic strokes. To broaden the scope of our introduction, we will include information about hemorrhagic events and cardiovascular abnormalities associated with ET.

  1. Insufficient rationale: While the introduction states that the study aims to provide a detailed analysis of five patients with ET-associated ischemic strokes, it does not clearly explain why this analysis is necessary or how it will contribute to the existing knowledge in the field. Adding a clear rationale for conducting this specific study and highlighting any gaps in the current literature would enhance the introduction.

> We agree that our introduction could benefit from a clearer explanation of the necessity of our analysis and its potential contribution to the field. Therefore, we plan to elaborate on why our detailed analysis of patients with ET-associated ischemic strokes is necessary, particularly how it aims to fill current knowledge gaps.

  1. Lack of objectives: The introduction does not explicitly state the objectives or research questions of the study. Including clear objectives would help readers understand the purpose and focus of the research.

> We understand the importance of explicitly stating the study objectives in the introduction. We will revise our introduction to include clear research objectives to better guide the readers through our study.

Methods:

  1. The methods do not provide specific information about data collection procedures, the process of data extraction, how missing data were handled, and any statistical analyses performed. Including these details would enhance the transparency and reproducibility of the study.

> Thank you for your thoughtful comments and suggestions. We appreciate your attention to the transparency and reproducibility of our study. Our study focused on reviewing the records of 5 patients diagnosed with ET, and we made every effort to ensure that the data was as complete as possible. As for the statistical analyses, given that our study was largely descriptive, focusing on individual patient cases, we did not conduct statistical analyses. we revised our methods section as you recommended.

Results:

  1. What are the long-term recurrence rates of ischemic stroke in patients with JAK2 mutation-associated essential thrombocythemia (ET) and how do they compare to other patient populations?

> As we know, stroke recurrence rates are highly variable in the general population, with reported rates ranging from 5.7% to 51.3% depending on a variety of factors, including the etiology of the initial stroke, the patient's age, presence of comorbidities, and how effectively risk factors are managed.

In our patient cohort of 5 cases, despite one case discontinuing medication, the prognosis has generally been good. However, it's important to acknowledge the limitations of our sample size, which prevent us from making direct comparisons with larger, more generalized populations. Interestingly, we found a Japanese study ("Ischemic Stroke with Essential Thrombocythemia: A Case Series") that evaluated 11 ET patients among 2538 stroke patients. In this study, they reported a recurrence in 3 out of the 11 ET patients. While this provides valuable insight, the small number of ET patients again limits the generalizability of these results.

 We revised discussion section of the manuscript to mention this study as a reference point, which underlines the need for more extensive research on larger cohorts to better understand the recurrence rates and long-term outcomes of ischemic stroke in patients with ET.

  1. How does the combined antiplatelet and cytoreductive therapy influence outcomes and the maintenance of platelet counts below 600,000/μL in patients with JAK2 mutation-associated essential thrombocythemia (ET) and ischemic stroke.

> Thank you for your insightful comment. As discussed in our response to your earlier question regarding prognosis, our small cohort of patients showed generally favorable outcomes during a mean follow-up period of 3.5 ± 1.9 years. This outcome was observed under the treatment regimen of combined antiplatelet and cytoreductive therapy, which was instrumental in maintaining platelet counts below 600,000/μL. We have further discussed this aspect in our revised manuscript's discussion section.

Discussion:

  1. The discussion does not compare the outcomes or characteristics of the study's patients to a control group or other populations. A comparison could provide insights into the uniqueness or similarities of the observed findings compared to patients without ET or different etiologies of ischemic stroke.

> Thank you for your valuable suggestion. Our study was designed as a case series aimed at shedding light on the clinical features of ischemic stroke in patients with ET, and as such, we did not incorporate a control group for comparison. You correctly pointed out that the inclusion of a control group would have added depth to our analysis. However, due to the rarity of ET and the limited sample size of our study, we could not perform direct comparisons with other groups or populations. We concur with your sentiment that future studies involving larger cohorts would be beneficial to better understand and compare the characteristics and outcomes of ischemic stroke in ET patients with those of different etiologies.

  1. The discussion briefly explains the use of clopidogrel instead of aspirin for secondary stroke prevention in patients with confirmed JAK2 mutation and arterial thrombotic events. However, it does not discuss the rationale behind this decision or consider other possible treatment options. A more comprehensive analysis of the various therapeutic approaches available would provide a broader perspective on the management of ET-associated ischemic strokes.

> Thank you for the insightful comment. The choice of antiplatelet therapy for secondary prevention of stroke in patients with ET is indeed a critical aspect of the overall management strategy. Our study, as mentioned, deviated from current guidelines which recommend aspirin in combination with hydroxyurea for patients with a confirmed JAK2 mutation and arterial thrombotic events. In our cases, stroke was diagnosed before the identification of ET, leading to the initiation of clopidogrel treatment in the context of typical acute ischemic stroke management. Subsequent diagnosis of ET did not prompt a switch to aspirin, as our patients did not present with increased bleeding risk, and we found no compelling reason to alter the ongoing treatment regimen.

While current guidelines recommend aspirin for patients with ET and arterial thrombotic events, they do not offer guidance on other antiplatelet options. Furthermore, a lack of randomized controlled trials or expert opinions advocating for a specific antiplatelet therapy in ET patients leaves this area relatively unexplored. Therefore, we refrained from discussing this aspect in detail in our study. One patient in our series had clopidogrel switched to cilostazol, but this was not due to a specific problem or concern.

We agree with your suggestion that a comprehensive evaluation of the various therapeutic options available for secondary prevention of stroke in patients with ET is needed. Further research in larger cohorts could shed light on this topic, offering valuable insights into the optimal therapeutic strategy for these patients.

Conclusion:

Why doesn't the article have a conclusion section?

> Indeed, "Conclusion" section was integrated within the final paragraph of the discussion. Now, we revised the structure with a distinct conclusion.

Reviewer 2 Report

Dear authors, thank you to share with me this interesting paper. 

I HAVE SOME SUGGESTIONS TO IMPROUVE THE QUALITY OF YOUR WORK.

1) The Essential thrombocythemia is a very rare disease that only in some cases complicated with stroke/TIA. I suggest you to add in the introdution the incidence and prevalence rate of ET in the world and in tour country (5 cases in 9 years on a total of 961 patients is a very high rate). Add also the correct reference about that. 

2) In the table numb.1 add also the NIHSS and mRS ath the admission

3) Explain why the patient numb 1, as reported in table numb 1, switched from clopidogrel to cilostazol. Adverse events was occurred? Other?

4) in the figure numb 1 add the correct MRI sequences (DWI, TOF, FLAIR)

5) add short conclusions after discussion

Author Response

I HAVE SOME SUGGESTIONS TO IMPROUVE THE QUALITY OF YOUR WORK.

1) The Essential thrombocythemia is a very rare disease that only in some cases complicated with stroke/TIA. I suggest you to add in the introduction the incidence and prevalence rate of ET in the world and in your country (5 cases in 9 years on a total of 961 patients is a very high rate). Add also the correct reference about that.

> We appreciate your comment about the need to provide more context on the prevalence or significance of ET. Additionally, we have included a comparison with a study from Japan that reported a similar stroke rate in ET patients.

2) In the table numb.1 add also the NIHSS and mRS at the admission

> Thank you for your valuable suggestion. We added mRS at the admission (initial).

3) Explain why the patient numb 1, as reported in table numb 1, switched from clopidogrel to cilostazol. Adverse events was occurred? Other?

> Thank you for your insightful question. The decision to switch from clopidogrel to cilostazol in patient number 1 was not due to the occurrence of adverse events or any particular guidelines. Rather, the change was made based on the patient's consistent stable condition without any recurrence of stroke. It's important to note that current guidelines recommend aspirin for patients with ET and arterial thrombotic events but do not provide specific guidance on the use of other antiplatelet options. The lack of randomized controlled trials or expert consensus advocating for a particular antiplatelet therapy in patients with ET leaves this aspect open for further exploration and research. We hope this answers your query, and we appreciate your interest in our study.

4) in the figure numb 1 add the correct MRI sequences (DWI, TOF, FLAIR)

> Thank you for your suggestion. We understand the need for greater clarity in labeling the MRI sequences in Figure 1. Following your recommendation, we will add the correct MRI sequence labels to the figure. The images marked with † represent Fluid Attenuated Inversion Recovery (FLAIR) images, those marked with * represent Time of Flight (TOF) Magnetic Resonance Angiography (MRA), and those marked with ** indicate Computed Tomography Angiography (CTA). The unmarked images are Diffusion Weighted Imaging (DWI). We appreciate your attention to detail and your contribution to improving the quality of our work.

5) add short conclusions after discussion

> Indeed, "Conclusion" section was integrated within the final paragraph of the discussion. Now, we revised the structure with a distinct conclusion.